# Immediate Effects of Acupuncture on Explosive Force Production and Stiffness in Male Knee Joint

**DOI:** 10.3390/ijerph18189518

**Published:** 2021-09-09

**Authors:** Jun Wang, I-Lin Wang, Rui Hu, Shun Yao, Yu Su, Shu Zhou, Che-Hsiu Chen

**Affiliations:** 1Graduate Institute, Jilin Sports University, Freedom Road, No. 2476, Nanguan District, Changchun 130022, China; wjun5980@gmail.com (J.W.); ruihu0614@gmail.com (R.H.); yaoshun0330@gmail.com (S.Y.); y.su825@gmail.com (Y.S.); 2College of Physical Education, Hubei Normal University, Huangshi 435002, China; ilin@gms.ndhu.edu.tw; 3Department of Sport Performance, National Taiwan University of Sport, Taichung City 41354, Taiwan

**Keywords:** sham acupuncture, acupoints, isokinetic test, PAP

## Abstract

Acupuncture can improve explosive force production and affect joint stiffness by affecting muscle activation levels. This study aims to explore the effects of true acupuncture (TA) compared with sham acupuncture (SA) on the explosive force production and stiffness of the knee joint in healthy male subjects. Twenty subjects were randomly divided into the TA group (*n* = 10) and SA group (*n* = 10) to complete isokinetic movement of the right knee joint at a speed of 240°/s before and after acupuncture. Futu (ST32), Liangqiu (ST34), Zusanli (ST36), Xuehai (SP10), and Chengshan (BL57) were selected for acupuncture. The intervention of SA is that needles with a blunt tip were pushed against the skin, giving an illusion of insertion. The results showed that acupuncture and the intervention time had a significant interaction effect on knee joint explosive force and joint stiffness (*p* < 0.05). The average maximum (max) torque, average work, average power, average peak power and total work of the TA group increased significantly after acupuncture (*p* < 0.05), while the SA group did not (*p* > 0.05). Therefore, true acupuncture can immediately improve the explosive force and joint stiffness of the male knee joint by inducing post-activation potentiation (PAP) and/or De-Qi.

## 1. Introduction

In recent years, acupuncture as a nondrug therapy has been widely recognized and studied worldwide [1]. Traditional acupuncture therapy is also becoming increasingly popular in Western countries. In the UK, approximately 4 million acupuncture treatments were provided to patients in 2009 [2]. Acupuncture has been widely used in clinical medicine for the treatment of dysfunction caused by knee joint diseases. Past studies have shown that acupuncture improves balance function by enhancing the activity of brain neurons to increase the muscle strength of knee extensor muscle and hip flexor muscle of paralyzed and nonparalyzed limbs in stroke patients [3]. In early rehabilitation after total knee replacement, acupuncture treatment can alleviate the pain and swelling of the knee joint and accelerate the recovery of joint range of motion (ROM) [4]. The American College of Rheumatology (ACR) also suggests that acupuncture should be used for the treatment of patients who cannot undergo knee replacement [5]. Therefore, acupuncture is one of the commonly used treatments for pain and dysfunction caused by musculoskeletal diseases. In addition, acupuncture treatment can improve the walking speed and knee flexion moment of patients with knee osteoarthritis (OA), which has an immediate effect on transforming the gait control strategy into a normal mode [6]. Acupuncture can effectively improve the strength of the quadriceps femoris of athletes in sports competitions, which is conducive to improving sports performance and restoring neuromuscular function [7]. Therefore, acupuncture is an alternative therapy useful for improving sports performance and competition results in sports medicine.

Post-activation potentiation (PAP) can be defined as the acute enhancement of muscle contraction characteristics induced by maximum voluntary contraction (MVC) or submaximal voluntary muscle contraction [8]. PAP in humans may be induced by isometric maximum voluntary contraction, high-intensity resistance stimulus, and plyometric exercise [9,10,11]. There are two common mechanisms of the PAP effect. One is that phosphorylation of the myosin light chain caused by initial muscle activity makes actin and myosin molecules more sensitive to the availability of Ca^2+^, which increases the speed of strength development. The other is that the excitability of α-motor neurons increases contractility after muscle activity [12]. In addition, De-Qi is a mixture of aching, pressure, soreness, heaviness, fullness, warmth, cooling, numbness, tingling and dull pain [13]. The transmission of Qi involves rapid transmission by Aβ myelinated fibers with a higher threshold and slow conduction by unmyelinated C fibers with a lower threshold, but the deep muscle layer is rich in fibers with slow conduction speeds [14]. Stimulation of nerves can cause muscles to recruit additional motor units and increase the discharge rate of activated motor units in the De-Qi response induced by acupuncture treatment [15]. The insertion and operation of metal needles change the sensory input into the central nervous system and increase the excitability of α motor neurons [16]. Therefore, the possible mechanism of the De-Qi response induced by acupuncture is the enhancement of neuromuscular activity through increased motor unit recruitment and α motor neuron excitability. The PAP effect and the De-Qi caused by acupuncture may activate muscles by stimulating nerves, thus improving muscle strength and other sports performance. Past studies have proven that the PAP effect can effectively improve the explosive force of sprinters and healthy female shoulder joints [17,18]. Therefore, it may be possible to apply the principle that acupuncture can induce the PAP effect and/or De-Qi phenomenon to enhance the effect of acupuncture in improving the explosive force of the knee joint.

Explosive force refers to the ability to increase contraction torque as quickly as possible from a lower or static level under the nerve drive of the central nervous system and inherent muscle characteristics [19,20]. Acupuncture can increase the recruitment of motor units by stimulating nerve fibers at the acupoints and improves the coordination within and between muscles, thus increasing the ability of muscles to produce maximum power [21,22]. Accordingly, acupuncture has been used to improve the explosive force and speed ability of male lower limbs [23], and the strength, total work output, and explosive force output of quadriceps femoris can be improved combined with Tuina therapy [24]. Therefore, acupuncture may stimulate nerves to recruit more motor units and increase muscle strength production.

In sports such as sprinting, jumping and boxing and in situations where rapid joint stabilization is needed to avoid movement damage, the human ability to generate explosive force is considered important [25,26]. After 8 weeks of explosive training, the 30 m sprint time of young long-distance runners was significantly improved [27]. After explosive training is added to regular football training, the jumping ability of young football players has also been greatly improved [28]. Therefore, it is necessary to improve the ability to produce explosive force to prevent joint and fall injuries or to improve athletic performance. Additionally, previous studies have shown that the best athletic performance requires a certain stiffness. Too little joint stiffness can lead to excessive joint motion and soft tissue injury, while too much stiffness makes one more likely to suffer from bone injuries such as knee osteoarthritis and stress fracture [29]. Therefore, the purpose of this study was to explore the effects of TA on the generation of explosive force and stiffness of the knee joint in healthy male nonathlete subjects compared with SA. It is hypothesized that TA at specific points can improve the generation of knee joint explosive force and appropriately increase the stiffness of the knee joint to prevent joint injury through related neurophysiological reactions.

## 2. Materials and Methods

### 2.1. Subjects

Twenty male subjects from the Jilin Sports University of Health Technology College were selected to participate in this study. Through the health survey, all subjects were required to have no history of lower limb injury in the past 6 months. Subjects randomly allocated to the TA group (*n* = 10) received true acupuncture and the SA group (*n* = 10) received sham acupuncture. The participants were university students who were physically active but not athletes, and basic characteristics of subjects include age (23.8 ± 1.4 years), height: (1.7 ± 0.1 m), weight (66.4 ± 8.5 kg), and body mass index (BMI, 22.5 ± 2.9 kg/m^2^); these did not significantly differ before and after testing. The inclusion criteria for participation were that participants had no musculoskeletal or neurological disorders, had not been involved in regular strength training or structured sport training during the 6 months prior to the study, and had no previous experience with acupuncture or electrical stimulation [30]. All participants were right-leg dominant, as identified using an established questionnaire. Participants in the TA and SA groups were required to avoid any form of physical exercise and maintain a normal level of physical activity at least 48 h before the day. The procedures, purpose, and risks associated with the study were explained to all the subjects before they gave their written informed consent to participate.

### 2.2. Study Design and Protocol

The study adopted a randomized controlled trial and was approved by the Joint Institutional Review Board of Jilin Sport University (JLSU-IRB no. 2021002). After the subjects were recruited, they were assessed for eligibility to determine whether they meet the inclusion criteria. Then, the subjects who met the criteria were allocated to groups (TA group, *n* = 10; SA group, *n* = 10). During the formal experiment, the subjects in the TA and SA group mainly completed the following steps: pretest (before the acupuncture intervention), acupuncture, and posttest (after the acupuncture intervention). Posttests were carried out after 20 min of acupuncture treatment both in the TA and SA groups were assessed again for the same parameters to determine any change in the explosive force of the knee flexor and extensors. A flow diagram of the protocol is shown in Figure 1.

### 2.3. True Acupuncture and Sham Acupuncture Intervention

The TA and SA acupoints were chosen based on the concept of treatment to facilitate the systematic delivery of standardized acupuncture treatments. The following classical acupuncture points were used: ST32, ST34, ST36, SP10, and BL57 (Figure 2), which are located on the lower thighs and shanks. For the TA group, sterile, single-use disposable stainless steel needles (0.25 mm × 40 mm, Suzhou Medical Appliance Factory, Suzhou, People’s Republic of China) were inserted at traditional depths and angles on the unilateral right lower limb. Needles were inserted perpendicularly to a depth of 10–15 mm depending on the subjects’ constitution (e.g., skin thickness and subcutaneous fat layer thickness) and manipulated (bidirectionally rotated) until De-Qi arrival. The needle was left in place for 20 min while subjects rested in a sitting position. For SA group, which was a control for acupoint specificity, body locations not recognized as true acupoints or meridians for needling (sham acupuncture) were used [31]. The intervention mode of SA is that needles with a blunt tip were pushed against the skin for 20 min as in TA group, giving an illusion of insertion. The acupuncture treatments were performed by an experienced acupuncturist (JW) possessing a Health Professional Qualification certificate, which is approved and issued by the Ministry of Human Resources and Social Security of the People’s Republic of China and the National Health Commission. The same protocol was used in the TA and SA groups; thus, all criteria for harnessing nonspecific effects were included (same contact time and interaction between therapist and patient, manual contact during search for acupuncture points, and stimulation of the needles).

### 2.4. Measurement of Isokinetic Parameters

Assessment of explosive force performance included the average max torque/kg (Nm/kg), the average work/kg (J/kg), the average power/kg (W/kg), the average peak power/kg (W/kg), the total work (J), and joint stiffness of knee flexion (KF) and knee extension (KE). The strength testing was performed using an isokinetic training system (Con-Trex MJ; CMV AG, Dübendorf, Switzerland). Subjects were secured by body straps and seated comfortably in the dynamometer chair with an angle of <110° between the alignment of the spine and the femur, and the knee and hip flexed at 90°. The axis of rotation of the device was aligned with the anatomical axis of the knee. All participants were asked to change into comfortable clothes, and the right leg was secured with an inelastic band. The seating position, knee, and hip joint angles were all recorded during the familiarization session and used during the pre- and posttests to eliminate differences in strength output due to the seating position. After a specific warm-up (6 min running on a treadmill at a speed of 8 km/h), each subject performed five repetitive isokinetic KF and KE on the right knee at 240° s^−1^. Participants were instructed to extend their knee “as fast and hard as possible” for 1 s upon hearing an auditory signal, with the emphasis on “fast”. All isokinetic assessments started at knee angles of 90° and ~180° (full extension) for the KF and KE. As the arm of the dynamometer moved up from 90° to 180°, subjects were encouraged to perform maximally for each contraction throughout the full range of motion. The participants were verbally instructed to “push” or “pull” “as fast as possible”. The subjects relaxed as the dynamometer arm moved back to 90° (the passive phase of the contraction cycle).

### 2.5. Reasons for Acupoints Selection

According to traditional Chinese medicine (TCM), there is a network of meridians (jingluo) connecting functional organs in the human body. Acupuncture at specific acupoints along the meridians exerts therapeutic effects on nearby and/or distant regions [32]. For example, acupuncture Zusanli (ST 36) improved the level of strength and speed abilities in lower limb explosive strength in men [23]. In this study, ST32 and ST34 were located on the rectus femoris, SP10 on the medial femoris, ST 36 on the tibialis anterior, and BL57 on the gastrocnemius muscle. Acupuncture at these acupoints may have similar effects.

### 2.6. Data Analysis

Isokinetic parameters were further analyzed using MATLAB software (version R2019a; MathWorks, Inc., Natick, MA, USA) before and after acupuncture. The stiffness of the knee joint was calculated using the following formula:(1)$$K_{knee}=\frac{∆M_{knee}}{∆\theta_{knee}}$$
where the change in the joint moment between maximum knee flexion and maximum knee extension is defined as $∆M_{knee}$ and the change in the joint angle between maximum knee flexion and maximum knee extension is $∆\theta_{knee}$. The joint moment was normalized to the participant’s body weight.

### 2.7. Statistical Analysis

Statistical analyses were performed using MATLAB software (version R2019a; MathWorks, Inc., Natick, MA, USA). Descriptive and outcome statistics are presented as the means ± SD. The values of the average max torque for extension/flexion, average work for extension/flexion, average power for extension/flexion, average peak power for extension/flexion, total work for extension/flexion, and stiffness for extension/flexion were recorded before and after acupuncture intervention. There were between-subject factor group (TA group vs. SA group) and within-subject factor (pre- and posttest). This design allowed for testing the main effect of groups, the main effect of time, and the interaction of groups by time. Two-way analysis of variance (ANOVA) was used, considering the timepoint (pre- or posttest) as a within-subject effect and the group (TA and SA group) as a between-subject effect. In case of significant interaction, simple main effects were examined; that is, the effects of one factor, holding the other factor fixed. A *p*-value < 0.05 was considered statistically significant. When the interaction effect was significant, independent *t*-tests and paired *t*-tests were used to compare each variable between the pretest and posttest, both in the TA and SA groups. The effect sizes (ES) were measured by the Cohen’s d (Cohen’s d = Mean_post_−Mean_pre_/S.D.) and considers ES values as small (ES: 0.2–0.5), moderate (ES: 0.5–0.8), and large (ES: >0.8) [33].

## 3. Results

All participants completed the study procedures and did not report any adverse reactions or withdraw from the experiment. Table 1 shows the main effects (Time), main effects (Group), and interaction (Time * Group) in terms of the average max torque, average work, average power, average peak power, total work, and stiffness of knee joint flexion. The data showed that there was a significant difference in the interaction effects (group*times) (*p* < 0.01). Therefore, this suggests that between-subject factor (TA and SA group) and within-subject factor (pre- and posttest) significantly affect the explosive force and joint stiffness of knee joint flexion. Figure 3 shows the simple main effects of the TA group at pre- and posttest, and TA group and SA group at posttest. Specifically, TA group’s the average max torque Flex (*p* = 0.001, ES = 1.54), average work Flex (*p* < 0.001, ES = 1.81), average power Flex (*p* < 0.001, ES = 1.70), average peak power Flex (*p* = 0.002, ES = 1.38), and total work Flex (*p* = 0.002, ES = 1.40) significantly increased after acupuncture. In the SA group, there was no significant difference in those same parameters after acupuncture (*p* > 0.05). In addition, after acupuncture intervention, there were also significant differences between the TA group and the SA group, including the average maximum torque Flex (*p* < 0.001, ES = 2.10), the average work Flex (*p* < 0.001, ES = 1.96), and the average power Flex (*p* = 0.008, ES = 1.33), average peak power Flex (*p* = 0.001, ES = 1.88), and total work Flex (*p* = 0.010, ES = 1.28).

Table 2 shows the main effects (Time), main effects (Group), and interaction (Time * Group) in terms of the average max torque, average work, average power, average peak power, total work, and stiffness of knee joint extension. The data showed that there was a significant difference in the interaction effects (group*times) (*p* < 0.01). Therefore, this suggests that between-subject factor (TA and SA group) and within-subject factor (pre- and posttest) significantly affects the explosive force and joint stiffness of knee joint extension. Figure 4 shows the simple main effects of the TA group at pre- and posttest, and TA group and SA group at posttest. Specifically, the TA group’s the average max torque Ext (*p* = 0.001, ES = 1.45), average work Ext (*p* < 0.001, ES = 1.86), average power Ext (*p* = 0.009, ES = 1.06), average peak power Ext (*p* = 0.002, ES = 1.38) and total work Ext (*p* = 0.003, ES = 1.27) significantly increased after acupuncture. In the SA group, there was no significant difference in those same parameters after acupuncture (*p* > 0.05). Additionally, significant differences were also found in the TA group and the SA group after acupuncture intervention, including the average maximum torque Ext (*p* = 0.002, ES = 1.60), the average working Ext (*p* = 0.003, ES = 1.52), and the average power Ext (*p* = 0.012, ES = 1.25), average peak power Ext (*p* = 0.003, ES = 1.56), and total work Ext (*p* < 0.001, ES = 1.85).

Figure 5 shows the simple main effects of the joint stiffness in the TA group at pre-and posttest, and TA group and SA group at posttest. After acupuncture, significant differences were also found in the TA group and the SA group in terms of the stiffness Flex (*p* < 0.010, ES = 3.22) and stiffness Ext (*p* < 0.001, ES = 2.47). In the TA group, the stiffness Flex (*p* = 0.002, ES = 1.41) and the stiffness Ext (*p* < 0.001, ES =1.71) significantly increased after acupuncture, but there was no significant difference in the SA group (*p* > 0.05).

Taken together, our results suggest that 20 min of acupuncture at ST32, ST34, ST36, SP10, and BL57 can immediately improve the explosive force and joint stiffness of knee joint flexion/extension compared to sham acupuncture.

## 4. Discussion

The result of this study shows that after acupuncture at the specific acupoints ST32, ST34, ST36, SP10, and BL57 around the knee joint for 20 min, the isokinetic parameter values of posttest, including the average max torque, average work, average power, average peak power, and total work for flexion/extension and stiffness for flexion/extension, increase compared with pretest. These results indicate that true acupuncture can immediately improve the explosive force and joint stiffness of the knee joint compared to sham acupuncture.

The average max torque for flexion/extension and total work for flexion/extension of the knee joint increased after acupuncture in TA group. Previous studies have shown that acupuncture regulates the conduction velocity of muscle fibers by stimulating nerves [34]; the muscle can recruit more motor units and increase the firing rate of activated motor units when the nerve is stimulated [35]. Therefore, acupuncture may increase the contraction and conduction velocity of muscle fibers and thereby increase the average max torque and total work for flexion/extension of the knee joint in this study. Additionally, the PAP effect may be induced when nerve stimulation increases the recruitment of high-speed motor neurons [36]. The neuromuscular phenomenon caused by the PAP effect can increase the speed of force development and the max power and torque, which may be the result of the enhancement of myosin phosphorylation [37]. Therefore, in this study, acupuncture-induced muscle contraction caused by phosphorylation of myosin drives the PAP effect, which may be another reason for the increase in knee joint torque and total work after acupuncture.

The average work, average power, and average peak power for flexion/extension of the knee joint increased after acupuncture in TA group. Previous studies have shown that acupuncture at Zusanli (ST36) point of the lower extremity can produce a De-Qi sensation and activate different nerve fibers to affect the excitability of the motor cortex, improving the strength of knee extension and flexion [38,39,40]. The generation of De-Qi after acupuncture can effectively improve the pain and motor function of patients with osteoarthritis of the knee and improve the maximum peak torque and average power of athletes, thus improving sports performance, such as the rapid strength of the lower extremity [41,42]. Therefore, in this study, acupuncture at ST36 may induce the feeling of De-Qi and increase the work and power of the knee joint. In addition, the generation of muscle strength depends on the ability of the nervous system to effectively activate muscles through the recruitment of motor units, and the regulation of motor neuron activity is completed through spinal cord and spinal cord input [43]. Cumulative evidence indicates that inputs during acupuncture may activate various supraspinal regions in the central nervous system (CNS) [44], and the neurophysiologic measure of H-reflex indicated a significant increase of spinal motoneuron excitability after verum acupuncture [45]. Therefore, the increase in knee joint work and power after true acupuncture in our study may also be due to the increased excitability of spinal cord motor neurons after acupuncture.

The stiffness for flexion/extension of the knee joint increased after acupuncture in TA group. Past research has shown that through reflective or autonomy methods, knee joint muscle contraction can increase joint stiffness. Passive strain of the tissues around the knee joint and a change in its afferent innervation can also cause an increase in joint stiffness, which may change the regulation of motor neuron excitability according to a change in the pre-activation level [46]. The increase in the pre-activation level provides extra excitatory input for the motor pool and leads to more motor neuron activation [47]. Acupuncture can increase the recruitment of motor units and induce the nervous system to increase muscle activity by stimulating nerves [35]. Therefore, in this study, the reason for the increased joint stiffness of knee flexion/extension after acupuncture may be that acupuncture stimulated nerves increase the excitability of motor neurons and cause muscle contraction.

Nevertheless, our study possesses several limitations. First, although the five acupoints selected in the current study were stimulated in a random order, we cannot completely rule out the possible interaction between acupuncture and tactile stimulation or stimulation of different acupoints. Second, there was no measurement of muscle blood flow to determine the mechanism of acupuncture. In future research, it will be necessary to detect the change in muscle blood flow and explore the mechanism through which these changes are influenced by acupuncture.

## 5. Conclusions

In summary, compared with sham acupuncture, the average maximum torque, average work, average power, average peak power, total work of flexion/extension, and stiffness of flexion/extension increased after true acupuncture. It may be that after 20 min acupuncture at ST32, ST34, ST36, SP10 and BL57, PAP and/or the” De-Qi” phenomenon can be generated to improve the explosive force and joint stiffness of the knee joint after acupuncture. It demonstrated that application of acupuncture in sports training has a reliable basis, and it can be recommended that athletes combine acupuncture and explosive training in future research to improve the explosive performance of lower limbs.

## Figures and Tables

**Figure 1 ijerph-18-09518-f001:**
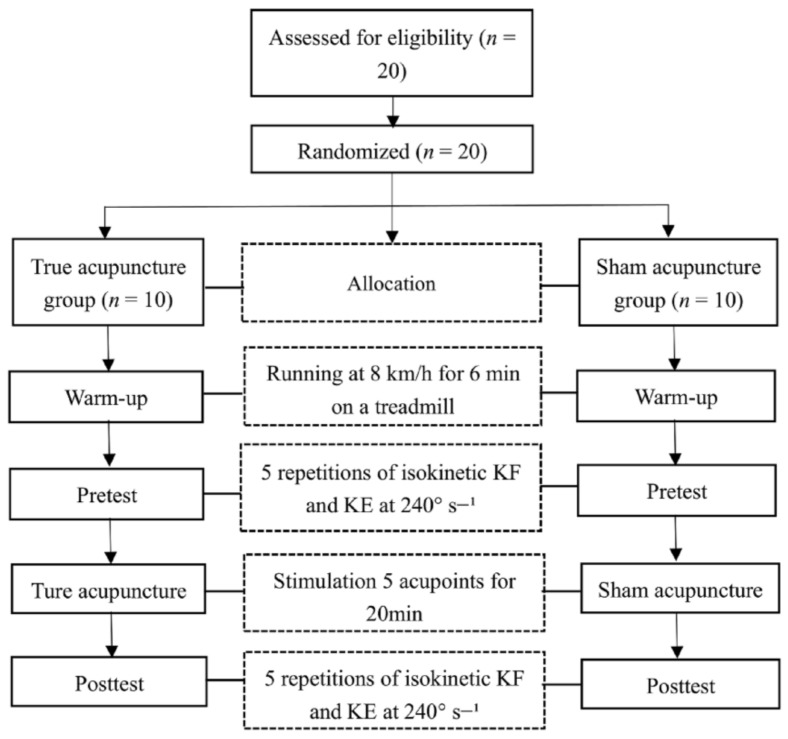
Flow diagram for the study. KF = knee flexion; KE = knee extension.

**Figure 2 ijerph-18-09518-f002:**
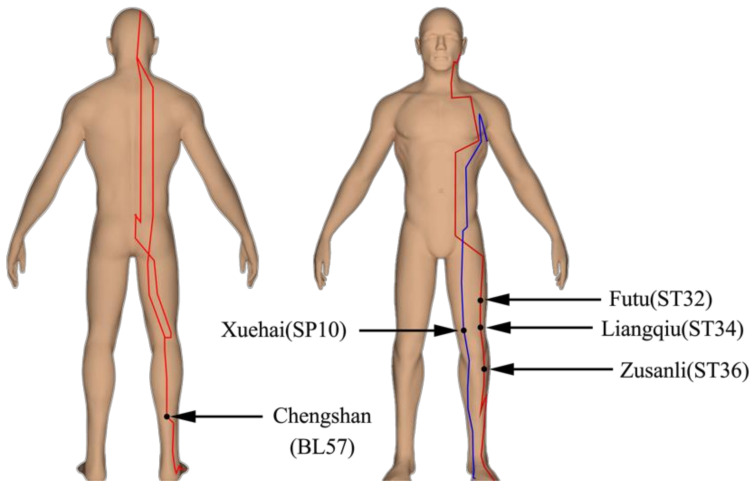
Schematic diagram of 5 acupuncture points located on the front and back sides of the lower extremity for true acupuncture and sham acupuncture.

**Figure 3 ijerph-18-09518-f003:**
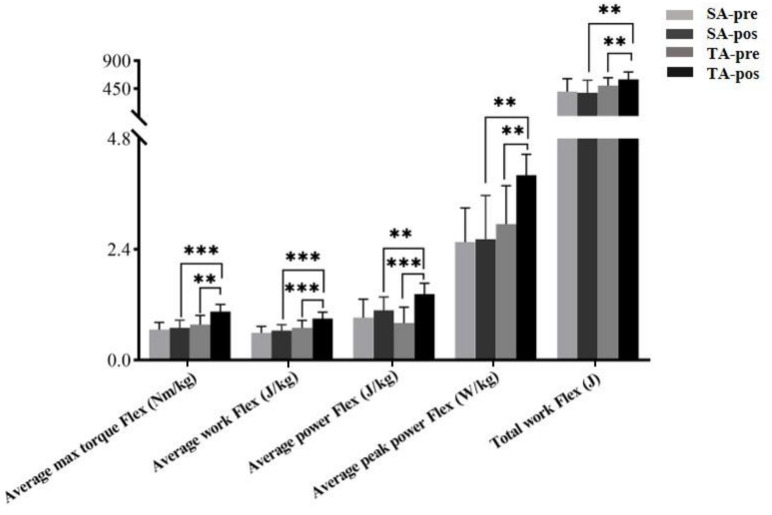
The simple main effects on knee flexion (Flex) before and after acupuncture for each isokinetic parameter in the SA group and TA group. True acupuncture group (TA), sham acupuncture group (SA). ** *p* < 0.01; *** *p* < 0.001.

**Figure 4 ijerph-18-09518-f004:**
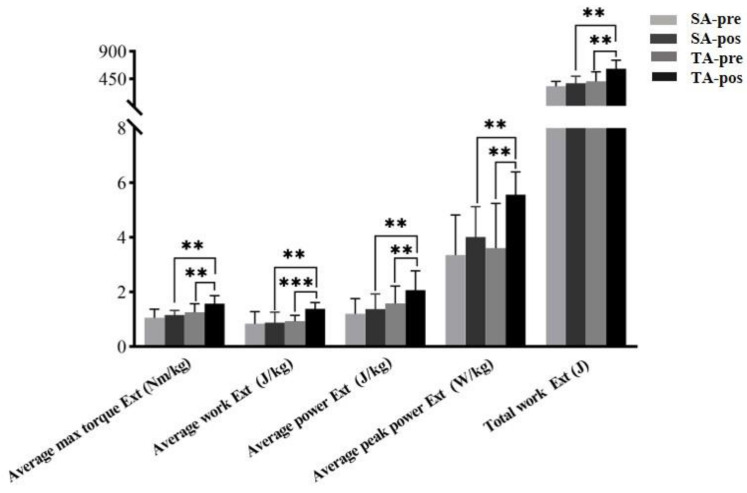
The simple main effects on knee extension (Ext) before and after acupuncture for each isokinetic parameter in the SA group and TA group. True acupuncture group (TA), sham acupuncture group (SA). ** *p* < 0.01; *** *p* < 0.001.

**Figure 5 ijerph-18-09518-f005:**
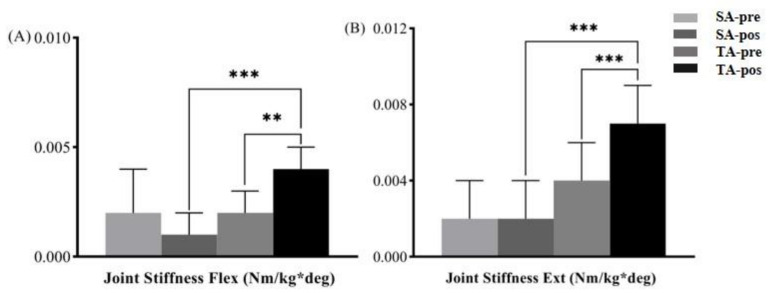
The simple main effects of knee joint stiffness for flexion (Flex) (**A**) and extension (Ext) (**B**) before and after acupuncture in the SA and TA groups. True acupuncture group (TA), sham acupuncture group (SA). ** *p* < 0.01; *** *p* < 0.001.

**Table 1 ijerph-18-09518-t001:** Mean ± SD of each variable for knee flexion (Flex) before and after acupuncture in TA and SA group.

Characteristic	Treatment	Pre	Post	*p* Values		
				Main Effects (*Time*)	Main Effects (*Group*)	Interaction (*Time* × *Group*)
Average max torque Flex/kg (Nm/kg)	SA	0.66 ± 0.16	0.70 ± 0.17	<0.001	0.004	0.002 *
	TA	0.77 ± 0.20	1.05 ± 0.16			
Average work Flex/kg (J/kg)	SA	0.58 ± 0.14	0.64 ± 0.13	<0.001	0.005	0.008 *
	TA	0.71 ± 0.16	0.90 ± 0.14			
Average power Flex/kg (J/kg)	SA	0.92 ± 0.40	1.08 ± 0.29	<0.001	0.365	0.008 *
	TA	0.80 ± 0.35	1.43 ± 0.24			
Average peak power Flex/kg (W/kg)	SA	2.56 ± 0.74	2.62 ± 0.95	0.015	0.024	0.027 *
	TA	2.95 ± 0.83	4.01 ± 0.45			
Total work Flex/kg (J)	SA	401.63 ± 212.40	384.41 ± 204.35	0.010	0.052	0.001 *
	TA	499.96 ± 129.86	600.44 ± 121.66			
Stiffness Flex/kg (Nm/Kg*deg)	SA	0.002 ± 0.002	0.001 ± 0.001	0.015	<0.001	0.001 *
	TA	0.002 ± 0.001	0.004 ± 0.001			

Values are expressed as the mean ± SD. True acupuncture group (TA), sham acupuncture group (SA). * There were significant differences in the interaction between Group and Time (*p* < 0.05).

**Table 2 ijerph-18-09518-t002:** Mean ± SD of each variable for knee extension (Ext) before and after acupuncture in TA and SA group.

Characteristic	Treatment	Pre	Post	*P* values		
				Main Effects (*Time*)	Main Effects (*Group*)	Interaction (*Time* × *Group*)
Average max torque Ext/kg (Nm/kg)	SA	1.06 ± 0.31	1.16 ± 0.22	<0.001	0.020	0.027 *
	TA	1.26 ± 0.31	1.57 ± 0.30			
Average work Ext/kg (J/kg)	SA	0.84 ± 0.44	0.88 ± 0.39	0.019	0.025	0.043 *
	TA	0.93 ± 0.22	1.38 ± 0.24			
Average power Ext/kg (J/kg)	SA	1.20 ± 0.56	1.37 ± 0.56	0.002	0.041	0.013 *
	TA	1.58 ± 0.64	2.07 ± 0.71			
Average peak power Ext/kg (W/kg)	SA	3.63 ± 1.46	4.01 ± 1.12	<0.001	0.178	0.008 *
	TA	3.61 ± 1.64	5.56 ± 0.84			
Total work Ext/kg (J)	SA	332.87 ± 78.60	378.72 ± 116.28	<0.001	0.003	0.021 *
	TA	410.89 ± 158.12	617.70 ± 140.73			
Stiffness Flex/kg (Nm/Kg*deg)	SA	0.002 ± 0.002	0.002 ± 0.002	0.002	<0.001	<0.001 *
	TA	0.004 ± 0.002	0.007 ± 0.002			

Values are expressed as the mean ± SD. True acupuncture group (TA), sham acupuncture group (SA). * There were significant differences in the interaction between Group and Time (*p* < 0.05).

## Data Availability

The data used to support the findings of this study are available from the corresponding author upon request.

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
