# Peer review of "Immediate Effects of Acupuncture on Explosive Force Production and Stiffness in Male Knee Joint"

_ijerph, 2021, doi:10.3390/ijerph18189518_

Round 1
Reviewer 1 Report
In the introduction the state-of-the-art in the field and the need for present study were adequately described. Experimental design and employed methods in the study are described in details. Results of the study are clearly presented and appropriate statistical tests were used.
However, I have a question regarding the Figure 1; Is there a mistake in the descriptions of Warm-up and Pretest? It would be logical that warm-up consisted of running while pretest should be the same as posttest.
Author Response
In the introduction the state-of-the-art in the field and the need for present study were adequately described. Experimental design and employed methods in the study are described in details. Results of the study are clearly presented and appropriate statistical tests were used.
However, I have a question regarding the Figure 1; Is there a mistake in the descriptions of Warm-up and Pretest? It would be logical that warm-up consisted of running while pretest should be the same as posttest.
RESPONSE: Thank you for your comment. The Figure 1 has been revised. (Please refer to line 134. Figure 1. Flow diagram for the study).

Reviewer 2 Report
Line 53. PAP not defined, please add.
Line 109. „no musculoskeletal or neurological disorders“ Please explain how it was determined? Clinical examination? History data?
Line 113. „maintain their normal level of physical activity during the entire study period“. Since the study was performed during a single day, please explain the term „study period“.
Line 119-121. The described main steps of the experiment do not match the central part of Figure 1. Please rewrite.
Line 151. Please change the figure caption to be clear enough to understand it without reading the whole paper.
Line 214. Relocate „At the end of the 20-min acupuncture, both the TA and SA groups were assessed again for the same parameters.“ to M&M.
Line 216. Is the simple main effect the same as effect size? Please use the phrase consistently.
Line 225. Table 1./Table 2. Was the main effect (time) considered for all groups together? Would it not be more appropriate to do it on each group separately?
Line 270. Figure 4. missing.
Line 334. „It demonstrated that acupuncture intervention in sports training can be used to improve sports performance.“-this conclusion is not directly supported by the results of the study. Please rewrite.
Author Response
- Line 53. PAP not defined, please add.
RESPONSE: Thank you for your comment. The sentence has been revised. (Please refer to line 51-54).
Original sentence: Post-activation potentiation can be defined as the acute enhancement of muscle contraction characteristics induced by maximum voluntary contraction (MVC) or submaximal voluntary muscle contraction (Baudry & Duchateau, 2007).
Altered sentence: Post-activation potentiation can be defined as the acute enhancement of muscle contraction characteristics induced by maximum voluntary contraction (MVC) or submaximal voluntary muscle contraction (Baudry & Duchateau, 2007). The PAP in humans may be induced by isometric maximum voluntary contraction, high-intensity resistance stimulus, and plyometric exercise (Hamada et al., 2000; Mcbride et al., 2005; Turner et al., 2015).
- Line 109. „no musculoskeletal or neurological disorders “Please explain how it was determined? Clinical examination? History data?
RESPONSE: Thank you for your comment. The sentence has been revised. (Please refer to line 104-106).
Original sentence: Twenty male subjects from the Jilin Institute of Physical Education Sports Health Technology College were selected to participate in this study.
Altered sentence: Twenty male subjects from the Jilin Institute of Physical Education Sports Health Technology College were selected to participate in this study. Through the health survey, all subjects were required to have no history of lower limb injury in the past 6 months.
- Line 113. „maintain their normal level of physical activity during the entire study period“. Since the study was performed during a single day, please explain the term „study period“.
RESPONSE: Thank you for your comment. The sentence has been revised. (Please refer to line 116-118).
Original sentence: Subjects were instructed to maintain their normal level of physical activity during the entire study period and to refrain from any form of physical exercise for at least 48 h prior to the tests.
Altered sentence: Participants in the TA and SA groups were required to avoid any form of physical exercise and maintain a normal level of physical activity at least 48 hours before the day.
- Line 119-121. The described main steps of the experiment do not match the central part of Figure 1. Please rewrite.
RESPONSE: Thank you for your comment. The main steps of the experiment have been rewritten. (Please refer to line 122-127).
Original sentence: The main steps of the experiment were carried out in the following order: warm-up, pretest (before the acupuncture intervention), acupuncture, and posttest (after the acupuncture intervention).
Altered sentence: After the subjects are recruited, they are assessed for eligibility to determine whether they meet the inclusion criteria. Then, the subjects who meet the criteria are allocated to groups (true acupuncture group, n=10; sham acupuncture group, n=10). During the formal experiment, the subjects in the TA group and SA group mainly completed the following steps: pretest (before the acupuncture intervention), acupuncture, and post-test (after the acupuncture intervention).
- Line 151. Please change the figure caption to be clear enough to understand it without reading the whole paper.
RESPONSE: Thank you for your comment. The figure captions have been changed. (Please refer to line 158-159).
Original figure caption: Figure 2. Acupuncture points.
Altered figure caption: Figure 2. Schematic diagram of 5 acupuncture points located on the front and back sides of the lower extremity for true acupuncture and sham acupuncture.
- Line 214. Relocate „At the end of the 20-min acupuncture, both the TA and SA groups were assessed again for the same parameters. “ to M&M.
RESPONSE: Thank you for your comment. The sentence has been relocated to M&M. (Please refer to line 128-130).
Original sentence: At the end of the 20-min acupuncture, both the TA and SA groups were assessed again for the same parameters.
Altered sentence: Posttests were carried out after 20 min of acupuncture treatment both in the TA and SA groups were assessed again for the same parameters to determine any change in the explosive force of the knee flexor and extensors.
- Line 216. Is the simple main effect the same as effect size? Please use the phrase consistently.
RESPONSE: Thank you for your comment. The simple main effect is not the effect size. The effect sizes (ES) measured by the Cohen's d (Cohen’s d = Meanpost−Meanpre/ S.D.) and considers ES values as small (ES: 0.2–0.5), moderate (ES: 0.5–0.8), and large (ES: >0.8) (Stival et al., 2014). (Please refer to line 212-214).
- Line 225. Table 1./Table 2. Was the main effect (time) considered for all groups together? Would it not be more appropriate to do it on each group separately?
RESPONSE: Thank you for your comment. The results section has been rewritten. (Please refer to line 216-286).
- Line 270. Figure 4. missing.
RESPONSE: Thank you for your comment. Figure 4 has been added. (Please refer to line 286, Figure 5).
- Line 334. „It demonstrated that acupuncture intervention in sports training can be used to improve sports performance. “-this conclusion is not directly supported by the results of the study. Please rewrite.
RESPONSE: Thank you for your comment. The sentence has been revised. (Please refer to line 356-363).
Original sentence: In summary, after TA, the average max torque, average work, average power, average peak power, and total work for flexion/extension and stiffness for flexion/extension in-creased compared with SA. The current study suggests that acupuncture as replacement therapy, acupuncture for 20-min at ST32, ST34, ST36, SP10 and BL57, could generate the effects of PAP or “De-Qi” to immediately improve non-athlete lower limb explosive force and joint stiffness. It demonstrated that acupuncture intervention in sports training can be used to improve sports performance. Future prospective studies should evaluate whether the acupuncture-induced increase in knee joint explosive force is timeliness, and its application in improving the actual performance of athletes.
Altered sentence: In summary, compared with sham acupuncture, the average maximum torque, aver-age work, average power, average peak power, total work of flexion/extension, and stiffness of flexion/extension increased after true acupuncture. It may be that after 20 min acupuncture at ST32, ST34, ST36, SP10 and BL57, PAP and (or) "De-Qi" phenome-non can be generated to improve the explosive force and joint stiffness of the knee joint after acupuncture. It demonstrated that application of acupuncture in sports training has a reliable basis, and it can be recommended that athletes combine acupuncture and explosive training in future research to improve the explosive performance of lower limbs.
Baudry, S., & Duchateau, J. (2007). Postactivation potentiation in a human muscle: effect on the rate of torque development of tetanic and voluntary isometric contractions. Journal of applied physiology, 102(4), 1394-1401.
Hamada, T., Sale, D. G., MacDougall, J. D., & Tarnopolsky, M. A. (2000). Postactivation potentiation, fiber type, and twitch contraction time in human knee extensor muscles. Journal of applied physiology.
Mcbride, J. M., Nimphius, S., & Erickson, T. M. (2005). The acute effects of heavy-load squats and loaded countermovement jumps on sprint performance. The Journal of Strength & Conditioning Research, 19(4), 893-897.
Stival, R. S. M., Cavalheiro, P. R., Stasiak, C., Galdino, D. T., Hoekstra, B. E., & Schafranski, M. D. (2014). Acupuncture in fibromyalgia: a randomized, controlled study addressing the immediate pain response. Revista Brasileira de Reumatologia (English Edition), 54(6), 431-436.
Turner, A. P., Bellhouse, S., Kilduff, L. P., & Russell, M. (2015). Postactivation potentiation of sprint acceleration performance using plyometric exercise. The Journal of Strength & Conditioning Research, 29(2), 343-350.

Reviewer 3 Report
- A brief summary
The article entitled “Immediate Effects of Acupuncture on Explosive Force Production and Stiffness in Male Knee Joint” by I-Lin Wang, Jun Wang, Rui Hu, Shun Yao, Yu Su, Shu Zhou, and Che-Hsiu Chen describes the changes in important functions of the muscles around the knee joint after acupuncture of certain acupoints
- Broad comments
The paper reads very well and is easy to understand.
But when the real measured data is presented, there is confusion about numbers in the text, numbers in the table and the columns in the graph.
I believe, there is an even significant effect, but the conclusions drawn are – to my opinion – going too far in some cases.
- Specific comments
Please find as comments in the text.
Please find comments in the PDF

Author Response
- Line 71-73. Which third effect of acupuncture/ the effect of acupuncture do you mean?
RESPONSE: Thank you for your comment. The sentence has been revised. (Please refer to line 73-75).
Original sentence: Therefore, acupuncture can be used to induce the PAP effect and De-Qi phenomenon, which may enhance the effect of acupuncture on improving the explosive force of the knee joint.
Altered sentence: Therefore, it may be possible to apply the principle that acupuncture can induce PAP effect and/or De-Qi phenomenon to enhance the effect of acupuncture in improving the explosive force of the knee joint.
- Line 138. how long did the blunt tip needles had contact to the skin?
RESPONSE: Thank you for your comment. The sentence has been revised. (Please refer to line 148-149).
Original sentence: The intervention mode of SA is that needles with a blunt tip were pushed against the skin, giving an illusion of insertion.
Altered sentence: The intervention mode of SA is that needles with a blunt tip were pushed against the skin for 20 minutes as in TA group, giving an illusion of insertion.
- Line 199. How did you check significance before you use t-tests?
RESPONSE: Thank you for your comment. The sentence has been revised. (Please refer to line 200-209).
Original sentence: The values of the average max torque for extension/flexion, average work for extension/flexion, average power for extension/flexion, average peak power for extension/flexion, total work for extension/flexion, and stiffness for extension/flexion observed after the treatment were compared between the groups using a mixed-design two-way repeated-measures analysis of variance (ANOVA) considering the timepoint (pre- or posttreatment) as a within-subject effect and the group (TA and SA group) as a between-subject effect.
Altered sentence: The values of the average max torque for extension/flexion, average work for extension/flexion, average power for extension/flexion, average peak power for extension/flexion, total work for extension/flexion, and stiffness for extension/flexion were recorded before and after acupuncture intervention. There were between-subject factor group (TA group vs. SA group) and within-subject factor (pre- and posttest). This design allowed for testing the main effect of groups, the main effect of time, and the interaction of groups by time. Using two-way analysis of variance (ANOVA) considering the timepoint (pre- or posttest) as a within-subject effect and the group (TA and SA group) as a between-subject effect. In case of significant interaction, simple main effects were examined, that is the effects of one factor holding the other factor fixed.
- Line 201-203. I don´t understand this sentence.
RESPONSE: Thank you for your comment. The sentence has been revised. (Please refer to line 212-214).
Original sentence: Effect size (ES) is an index that estimates can be used to compare treatment effects for different variables in the same study or for the same or different variables across different studies, regardless of the study sample size and the original scales of the variables. Calculation of the ES is simply the difference between the means of experimental (Me) and control (Mc) groups divided by the standard deviation for the control group (σc).
Altered sentence: The effect sizes (ES) measured by the Cohen's d (Cohen’s d = Meanpost−Meanpre/ S.D.) and considers ES values as small (ES: 0.2–0.5), moderate (ES: 0.5–0.8), and large (ES: >0.8) (Stival et al., 2014).
- Line 205. Who is this? Not in the literature mentioned
RESPONSE: Thank you for your comment. The sentence has been revised. (Please refer to line 212-214).
Original sentence: Cohen categorized ES values as small (ES: 0.2–0.5), moderate (ES: 0.5–0.8), and large (ES: >0.8) [30].
Altered sentence: The The effect sizes (ES) measured by the Cohen's d (Cohen’s d = Meanpost−Meanpre/ S.D.) and considers ES values as small (ES: 0.2–0.5), moderate (ES: 0.5–0.8), and large (ES: >0.8) (Stival et al., 2014).
- Figure 3. why do you split the axis in the lower part? why do you use the unevennumbers? What is shown - effects or mean?
RESPONSE: Thank you for your comment. The Figure 3 has been revised. (Please refer to line 245-246: Figure 3, line 272-273: Figure 4).
Original sentence: Figure 3. The simple effects on knee flexion (A) and extension (B) before and after acupuncture for each isokinetic parameter in the SA group and TA group
Altered sentence: Figure 3. The simple main effects on knee flexion (Flex) before and after acupuncture for each iso-kinetic parameter in the SA group and TA group. Figure 4. The simple main effects on knee extension (Ext) before and after acupuncture for each isokinetic parameter in the SA group and TA group
- Line 235. “Values are mean ±SD.” Which mean values do you mean? Not the ones from the table above.
RESPONSE: Thank you for your comment. The data (Mean ± SD) used for Figure 3, Figure 4 and Figure 5 comes from each isokinetic parameter in the SA group and TA group, and corresponds to the Mean±SD of each parameter in the Table 1 and Table 2.
- Line 332-333. you did not show that.
RESPONSE: Thank you for your comment. The sentence has been revised. (Please refer to line 349-366).
Original sentence: In summary, after TA, the average max torque, average work, average power, average peak power, and total work for flexion/extension and stiffness for flexion/extension increased compared with SA. The current study suggests that acupuncture as replacement therapy, acupuncture for 20-min at ST32, ST34, ST36, SP10 and BL57, could generate the effects of PAP or “De-Qi” to immediately improve non-athlete lower limb explosive force and joint stiffness. It demonstrated that acupuncture intervention in sports training can be used to improve sports performance. Future prospective studies should evaluate whether the acupuncture-induced increase in knee joint explosive force is timeliness, and its application in improving the actual performance of athletes.
Altered sentence: In summary, compared with sham acupuncture, the average maximum torque, average work, average power, average peak power, total work of flexion/extension, and stiffness of flexion/extension increased after true acupuncture. It may be that after 20 min acupuncture at ST32, ST34, ST36, SP10 and BL57, PAP and (or) "De Qi" phenomenon can be generated to improve the explosive force and joint stiffness of the knee joint after acupuncture. It demonstrated that application of acupuncture in sports training has a reliable basis, and it can be recommended that athletes combine acupuncture and explosive training in future research to improve the explosive performance of lower limbs.
- Line 283.Which receptor?
RESPONSE: Thank you for your comment. The sentence has been revised. (Please refer to line 299-304).
Original sentence: Previous studies have shown that acupuncture may cause the receptor to send nerve impulses to the spinal cord or act on the ascending pathway of the brain, resulting in the release of neurotransmitters and thus regulating the central function of the brain (Shen, 2001).
Altered sentence: Previous studies have shown that acupuncture regulates the conduction velocity of muscle fibers by stimulating nerves (Maioli et al., 2006), the muscle can recruit more motor units and increase the firing rate of activated motor units when the nerve is stimulated (Connelly & Vandervoort, 2000). Therefore, acupuncture may increase the contraction and conduction velocity of muscle fibers and thereby increase the average max torque and total work for flex-ion/extension of the knee joint in this study.
- Line 290-292. you did not show that.
RESPONSE: Thank you for your comment. The sentence has been revised. (Please refer to line 309-311).
Original sentence: Therefore, in this study, muscle contraction caused by acupuncture induced myosin phosphorylation, driving the PAP effect and increasing the knee extensor.
Altered sentence: Therefore, in this study, acupuncture induced muscle contraction caused by phosphorylation of myosin drives the PAP effect, which may be another reason for the increase in knee joint torque and total work after acupuncture.
- Line 305-307.what do you mean?
RESPONSE: Thank you for your comment. The sentence has been revised. (Please refer to line 324-328).
Original sentence: Acupuncture stimulation of somatosensory afferent nerves can induce short-term and long-term changes in neuromuscular excitability in the spinal cord and nerve levels above the spinal cord (Hübscher et al., 2010). Therefore, the increase in work and power of the knee joint in our study may also be due to the increased excitability of spinal motor neurons after acupuncture.
Altered sentence: Cumulative evidence indicates that inputs during acupuncture may activate various supraspinal regions in the central nervous system (CNS) (Napadow et al., 2005), and the neurophysiologic measure of H-reflex indicated a significant increase of spinal motoneuron excitability after verum acupuncture (Fink et al., 2004). Therefore, the increase in knee joint work and power after true acupuncture in our study may also be due to the increased excitability of spinal cord motor neurons after acupuncture.
Submission Date
30 July 2021
Date of this review
17 Aug 2021 11:02:57
Connelly, D. M., & Vandervoort, A. A. (2000). Effects of isokinetic strength training on concentric and eccentric torque development in the ankle dorsiflexors of older adults. The Journals of Gerontology Series A: Biological Sciences and Medical Sciences, 55(10), B465-B472.
Fink, M., Rollnik, J. D., Bijak, M., Borstädt, C., Däuper, J., Guergueltcheva, V., Dengler, R., & Karst, M. (2004). Needle acupuncture in chronic poststroke leg spasticity. Archives of physical medicine and rehabilitation, 85(4), 667-672.
Hübscher, M., Vogt, L., Ziebart, T., & Banzer, W. (2010). Immediate effects of acupuncture on strength performance: a randomized, controlled crossover trial. European journal of applied physiology, 110(2), 353-358. https://doi.org/10.1007/s00421-010-1510-y
Maioli, C., Falciati, L., Marangon, M., Perini, S., & Losio, A. (2006). Short- and long-term modulation of upper limb motor-evoked potentials induced by acupuncture. Eur J Neurosci, 23(7), 1931-1938. https://doi.org/10.1111/j.1460-9568.2006.04698.x
Napadow, V., Makris, N., Liu, J., Kettner, N. W., Kwong, K. K., & Hui, K. K. (2005). Effects of electroacupuncture versus manual acupuncture on the human brain as measured by fMRI. Human brain mapping, 24(3), 193-205.
Shen, J. (2001). Research on the neurophysiological mechanisms of acupuncture: review of selected studies and methodological issues. The Journal of Alternative and Complementary Medicine, 7(1), 121-127.
Stival, R. S. M., Cavalheiro, P. R., Stasiak, C., Galdino, D. T., Hoekstra, B. E., & Schafranski, M. D. (2014). Acupuncture in fibromyalgia: a randomized, controlled study addressing the immediate pain response. Revista Brasileira de Reumatologia (English Edition), 54(6), 431-436.
